# Antihypertensive treatment and risk of cardiovascular mortality in patients with chronic kidney disease diagnosed based on the presence of proteinuria and renal function: A large longitudinal study in Japan

Kei Nagai[1], Kunihiro Yamagata[1,2]*, Kunitoshi Iseki[2,3], Toshiki Moriyama[2,4], Kazuhiko Tsuruya[2,5], Shouichi Fujimoto[2,6], Ichiei Narita[2,7], Tsuneo Konta[2,8], Masahide Kondo[1,2], Masato Kasahara[2,9], Yugo Shibagaki[2,10], Koichi Asahi[2,11], Tsuyoshi Watanabe[2,12]

1 University of Tsukuba, Tsukuba, Ibaraki, Japan, 2 The Steering Committee for "Design of the Comprehensive Health Care System for Chronic Kidney Disease (CKD) Based on the Individual Risk Assessment by Specific Health Checkups", Tsukuba, Ibaraki, Japan, 3 Okinawa Heart and Renal Association, Okinawa, Japan, 4 Health Care Center, Osaka University, Suita, Japan, 5 Nara Medical University, Nara, Japan, 6 University of Miyazaki, Miyazaki, Japan, 7 Niigata University Graduate School of Medical and Dental Sciences, Niigata, Japan, 8 Yamagata University Graduate School of Medical Science, Yamagata, Japan, 9 Institute for Clinical and Translational Science, Nara Medical University Hospital, Nara, Japan, 10 St. Marianna University School of Medicine, Kawasaki, Kanagawa, Japan, 11 Iwate Medical University, Morioka, Japan, 12 Fukushima Rosai Hospital, Iwaki, Japan

* k-yamaga@md.tsukuba.ac.jp

**Data Availability Statement:** We used data available upon request due to ethical restrictions on

## Abstract

Several recent clinical trials and meta-analyses have shown that lowering blood pressure reduces the risk of cardiovascular disease. However, current evidence that describes general demographics in blood pressure and mortality with chronic kidney disease is sparse in Japan. Using a population-based longitudinal cohort that received annual health checkups in Japan in 2008, hypertensive status, self-reported use of antihypertensive drugs, and prognosis were examined through 2012. Chronic kidney disease was defined as positive proteinuria or estimated glomerular filtration rate <60 ml/min/1.73 m$^2$. Subjects were 40 to 74 years old (n = 227,204) with median 3.6 years follow-up period, and patients with and without chronic kidney disease were analyzed separately (n = 183,586 and n = 43,618, respectively). Cardiovascular disease mortality, comprising coronary heart diseases and stroke as entered in the national death registry using ICD-10 coding, was examined. Among all subjects, 346 deaths (96 in chronic kidney disease and 250 in non-chronic kidney disease) due to cardiovascular disease occurred. Compared with cardiovascular disease mortality in chronic kidney disease patients with untreated normal blood pressure, the multivariable adjusted hazard ratio was 3.08 (95% confidence interval: 1.75–5.41) for those with untreated hypertension, 2.30 (1.31–4.03) for those who became normotensive after treatment, and 3.28 (1.91–5.64) for those who remained hypertensive despite treatment. In non-chronic kidney disease subjects, the ratios were 1.90 (1.33–5.41), 1.95 (1.35–2.80), and 1.77 (1.18–2.66), respectively. These results from a nationwide cohort could be one of

sharing data publicly. The protocol of this project (Research on the Positioning of Chronic Kidney Disease in Specific Health Check and Guidance in Japan) determined that analytical data were distributed only to the members of steering committee to avoid any possibility that someone else identify individuals of this cohort. Because the data contain potentially identifying information (i.e. prefectural number and date of health checking), our institutional ethics committee has imposed them. Also, data had been obtained with the protocol approved by the relevant institutional ethical review board. We would like to put the information regarding data access committee; Department of Chronic Kidney Disease Initiatives Fukushima Medical University School of Medicine, Fukushima, Japan (dckdi@fmu.ac.jp).

**Funding:** KY was granted from the Japan Agency for Medical Research and Development (AMED) under Grant Number JP17ek0310005, JP18/JP19ek0310010 and 18lk1010033. (https://www.amed.go.jp/). The funders had no role in study design, data collection and analysis, decision to publish, or preparation of the manuscript.

**Competing interests:** The authors have declared that no competing interests exist.

representative demographics of controlling blood pressure and cardiovascular disease deaths when treating patients with chronic kidney disease in Japan in recent years. Even after development and spread of anti-hypertensive drugs, preventing development of hypertension is preferable, because any hypertension treatment status comparing untreated normal blood pressure was a risk of cardiovascular mortality at baseline year.

## Introduction

High blood pressure (BP) confers a risk of cardiovascular disease (CVD) [1–3]. Compared with the general population, patients with chronic kidney disease (CKD) have a high risk of CVD mortality [4]. Guidelines recommend lower BP targets in the CKD population than in the general population to slow the progression of CKD [5,6]. Antihypertensive therapy is well known to reduce CVD risk in the general population [7,8]. However, there is little evidence regarding BP treatment and CVD death in patients with CKD.

Numerous observational studies focused on general populations have found that, for a given baseline BP, the risk of CVD is higher for people who use antihypertensive medications than for those who do not [1,9–14]. However, most of them failed to consider participants' renal function and proteinuria, which are well-known determinants of cardiovascular risk, because the cohorts were established decades ago, before the current definitions of CKD were developed. A recent meta-analysis of data from randomized, controlled trials showed the effectiveness of BP reduction by antihypertensive medications [15]. However, systematic searches of 123 randomized trials demonstrated that only a few reported the renal function of the cohort. For example, of 31 BP-lowering trials assessing different BP targets, only 6 studies selected cohorts that included CKD patients [16–21], whereas 12 studies defined no-CKD cohorts, and the remaining 13 did not report the renal function of their cohorts. These results also indicate the rarity of longitudinal cohorts that include a well-identified CKD population for assessing CVD mortality.

The present study evaluated a longitudinal, general-population cohort of 227,204 persons who received annual health checkups, including examinations for proteinuria and renal function according to "The Specific Health Check and Guidance in Japan" program in 2008, and followed their prognosis and cause of death through December 2012. In subpopulations with and without CKD diagnosed based on the presence of proteinuria and renal function, risks for all-cause and CVD mortality were examined among various categories of BP control. This analysis provides information about the representative status of BP control and CVD mortality of Japanese people with CKD in recent years.

## Methods

### Patients and methods

This longitudinal cohort study was conducted according to the guidelines of the Declaration of Helsinki and was granted ethics approval by the relevant institutional review boards (University of Tsukuba for ethical issues approved as No. 999, UMIN: 000019774). Original Ethics Committee approval was obtained from Fukushima Medical University (IRB #1485, #2771).

The study was performed as part of the prospective ongoing "Research on the Positioning of Chronic Kidney Disease in Specific Health Check and Guidance (so-called "Tokutei-Kenshin") in Japan" project [22]. Other details, such as the participants' areas of residence, were

reported previously [23–25]. Outliers were deleted through winsorization; they accounted for 0.01%–0.1% of the total. The raw database was solely used and managed by the statistician, and the principal analyses to identify those who died among screened subjects were completed by March 2015 and recently reported [26]. Subsequent analyses were done using a standard analysis file (SAF) without any personal identifiers.

The duration of follow-up of the subjects was 1 to 4 years (2008 through 2012, median duration was 3.6 years). The net subject population comprised 227,204 people (59.0% [n = 134,103] were women) aged from 40 to 74 years and for whom all of the data necessary for our research purposes were available. The data included information about age, sex, body mass index (BMI), systolic BP, diastolic BP, smoking habit, use of antihypertensive, lipid-lowering, and hypoglycemic drugs (obtained by self-reported questionnaire), and the results of dipstick urinalyses for proteinuria and blood tests for glucose levels, creatinine concentration, and lipid status.

## Mortality surveillance

The underlying causes of death were coded according to ICD-10. Follow-up was conducted through December 2012. Incidents of CVD death were defined by ICD coding as I20-29 and I60-69.

## Measurement of parameters

Urinalysis by the dipstick method was performed on a single spot-urine specimen collected early in the morning. Urine dipstick results for proteinuria were interpreted by the medical staff at each local medical institution and recorded as −, +/−, 1+, 2+, and 3+. In Japan, the Japanese Committee for Clinical Laboratory Standards (http://jccls.org/) recommends that all urine dipstick results of 1+ correspond to a urinary protein level of 30 mg/dl; proteinuria was defined as 1+ or greater. Serum creatinine was measured using the enzymatic method. The glomerular filtration rate (GFR) was estimated using the formula of the Japanese Society of Nephrology [27]. CKD was defined as positive proteinuria or estimated GFR (eGFR) <60 ml/min/1.73 $m^2$. Hyperglycemia was defined as HbA1c ≥6.5%, and hypertension was defined as systolic BP ≥140 mmHg and diastolic BP ≥90 mmHg. Hypercholesterolemia was defined as low-density lipoprotein cholesterol ≥140 mg/dl, high-density lipoprotein cholesterol ≤40 mg/dl, or triglycerides ≥200 mg/dl. These comorbid conditions at the baseline year were used for the risk analysis.

## Statistical analysis

The primary outcomes for the analysis were all-cause and CVD deaths during the follow-up period. Variables were age, sex, HbA1c, hypertension, renal function, proteinuria, low-density lipoprotein-cholesterol, high-density lipoprotein-cholesterol, triglycerides, cigarette smoking, use of antihypertensive medication, use of lipid-lowering drugs, and treatment for diabetes. The hypertension treatment category was defined according to BP levels (normal, <140/<90 mmHg; hypertensive, ≥140/≥90 mmHg), as directed in the hypertension guidelines [28]. When the systolic and diastolic BPs were in different categories, the subject was assigned to the hypertension treatment category. After subjects were categorized according to BP, they were allocated to untreated normal, untreated hypertensive, treated normotensive, and treated hypertensive groups. Hazard ratios of the incidence of CVD were estimated using the Cox regression model (SAS version 9.4, SAS Institute, Cary, NC, USA). Other statistical analyses and graphical analyses were performed using Stata version 14 and GraphPad Prism version 6. A P value <0.05 was considered significant.

## Results

Table 1 presents the mean ages of the subjects and the means and proportions of risk factors at the baseline year, according to hypertension treatment category. Compared with subjects with untreated, normal BP at baseline, untreated hypertensive subjects were a mean of 2.4 years older, and the treated population was 5.5 years older on average; in addition, BMI was higher in hypertensive subjects, regardless of treatment category. Furthermore, eGFR was 1.3 ml/min/1.73 m$^2$ lower in the subpopulation with untreated hypertension and 5 ml/min/1.73 m$^2$ lower in subjects who remained hypertensive despite treatment, compared with that of subjects with untreated normal BP. The rate of positive proteinuria was higher in hypertensive than in normotensive subjects, and those with treated hypertension had the highest rate of positive proteinuria. The use of lipid-lowering drugs and of diabetes treatment paralleled that of hypertensive drugs. Finally, the rate of cigarette smoking was lower in treated than in untreated populations.

The characteristics of subjects with and without CKD are shown in Table 2. The trends in mean age, BMI, eGFR, and proportions of positive proteinuria, medication use, and cigarette smoking between subjects with and without CKD (Table 2) paralleled those between hypertension treatment categories (Table 1). Within a hypertension treatment category, subjects with CKD tended to include fewer women and have a higher mean age, higher BMI, worse dyslipidemia and hyperglycemia and a higher rate of positive proteinuria than did those without CKD (Table 2).

During follow-up, 2745 all-cause deaths (2107 non-CKD subjects and 638 CKD subjects) and 346 CVD deaths (250 non-CKD subjects and 96 CKD subjects) occurred in this cohort. The all-cause mortality and CVD mortality in subjects with and without CKD according to proteinuria and renal function are shown in Table 3. Dividing the number of all-cause

**Table 1. Study population.**

|  |  | Normotensive | Hypertensive | Normotensive | Hypertensive |  |
| --- | --- | --- | --- | --- | --- | --- |
|  |  | Untreated | Untreated | Treated | Treated | P |
| Number |  | 127,312 | 37,867 | 34,662 | 27,363 |  |
| Sex | %, women | 63.4 | 52.9 | 56.6 | 53.2 | <0.001 |
| Age | years | 60.4 ± 9.4 | 62.8 ± 8.2 | 65.9 ± 6.4 | 65.9 ± 6.5 | <0.001 |
| Height | cm | 157.2 ± 8.4 | 157.5 ± 8.8 | 156.0 ± 8.4 | 156.4 ± 8.6 | <0.001 |
| Weight | kg | 56.0 ± 10.0 | 59.3 ± 10.9 | 59.4 ± 10.4 | 60.9 ± 10.8 | <0.001 |
| Body mass index | kg/m$^2$ | 22.6 ± 3.1 | 23.8 ± 3.3 | 24.3 ± 3.3 | 24.8 ± 3.5 | <0.001 |
| Systolic blood pressure | mmHg | 118.9 ± 12.0 | 148.9 ± 12.8 | 126.2 ± 9.0 | 149.3 ± 11.5 | <0.001 |
| Diastolic blood pressure | mmHg | 71.7 ± 8.5 | 86.9 ± 9.8 | 74.7 ± 7.8 | 84.8 ± 9.5 | <0.001 |
| Triglycerides | mg/dl | 112 ± 76 | 132 ± 96 | 125 ± 76 | 133 ± 89 | <0.001 |
| High-density lipoprotein | mg/dl | 63 ± 16 | 61 ± 16 | 59 ± 15 | 59 ± 15 | <0.001 |
| Low-density lipoprotein | mg/dl | 126 ± 31 | 129 ± 32 | 120 ± 28 | 123 ± 29 | <0.001 |
| HbA1c | % | 5.3 ± 0.6 | 5.4 ± 0.8 | 5.5 ± 0.7 | 5.5 ± 0.8 | <0.001 |
| eGFR | ml/min/1.73 m$^2$ | 75.8 ± 15.5 | 74.5 ± 15.9 | 70.8 ± 16.2 | 71.0 ± 16.3 | <0.001 |
| Proteinuria | %, + or more | 3.2 | 6.7 | 7.8 | 11.2 | <0.001 |
| Use of antihypertensive drugs | %, yes | 0 | 0 | 100 | 100 | – |
| Lipid-lowering drug use | %, yes | 8.8 | 7.4 | 28.8 | 26.0 | <0.001 |
| Diabetes treatment | %, yes | 3.0 | 3.1 | 9.4 | 10.4 | <0.001 |
| Cigarette smoking | %, yes | 14.4 | 14.3 | 10.8 | 10.6 | <0.001 |

Low eGFR; less than 60 ml/min/1.73 m$^2$

**Table 2. Baseline characteristics of subpopulations with and without chronic kidney disease (CKD).**

| CKD (-) | | Normotensive Untreated | Hypertensive Untreated | Normotensive Treated | Hypertensive Treated | P |
|---|---|---|---|---|---|---|
| Number | | 108653 | 30341 | 25236 | 19356 | |
| Sex | %, women | 65.0 | 55.3 | 60.4 | 57.8 | <0.001 |
| Age | years | 59.7 ± 9.5 | 62.4 ± 8.3 | 65.5 ± 6.6 | 65.4 ± 6.7 | <0.001 |
| Height | cm | 157.0 ± 8.4 | 157.2 ± 8.8 | 155.6 ± 8.4 | 155.9 ± 8.5 | <0.001 |
| Weight | kg | 55.7 ± 10.0 | 58.8 ± 10.8 | 58.7 ± 10.2 | 60.1 ± 10.7 | <0.001 |
| Body mass index | kg/m$^2$ | 22.5 ± 3.1 | 23.7 ± 3.3 | 24.2 ± 3.3 | 24.6 ± 3.5 | <0.001 |
| Systolic blood pressure | mmHg | 119 ± 12 | 149 ± 13 | 126 ± 9 | 149 ± 11 | <0.001 |
| Diastolic blood pressure | mmHg | 72 ± 9 | 87 ± 10 | 75 ± 8 | 85 ± 9 | <0.001 |
| Triglycerides | mg/dl | 111 ± 75 | 130 ± 95 | 122 ± 74 | 129 ± 88 | <0.001 |
| High-density lipoprotein | mg/dl | 63 ± 16 | 62 ± 16 | 60 ± 15 | 60 ± 15 | <0.001 |
| Low-density lipoprotein | mg/dl | 126 ± 31 | 129 ± 32 | 120 ± 28 | 123 ± 29 | <0.001 |
| HbA1c | % | 5.3 ± 0.6 | 5.4 ± 0.7 | 5.4 ± 0.7 | 5.5 ± 0.7 | <0.001 |
| eGFR | ml/min/1.73 m$^2$ | 78.9 ± 13.9 | 78.1 ± 13.8 | 76.6 ± 13.1 | 76.8 ± 13.1 | <0.001 |
| Low eGFR | %, yes | 0 | 0 | 0 | 0 | - |
| Proteinuria | %, + or more | 0 | 0 | 0 | 0 | - |
| Use of antihypertensive drugs | %, yes | 0 | 0 | 100 | 100 | - |
| Use of lipid-lowering drug | %, yes | 8.4 | 7.2 | 28.0 | 25.7 | <0.001 |
| Diabetes treatment | %, yes | 2.8 | 2.9 | 8.5 | 9.0 | <0.001 |
| Cigarette smoking | %, yes | 14.8 | 14.3 | 10.8 | 10.4 | <0.001 |
| CKD (+) | | | | | | |
| Number | | 18659 | 7526 | 9426 | 8007 | |
| Sex | %, women | 53.9 | 43.0 | 46.4 | 41.8 | <0.001 |
| Age | years | 64.0 ± 7.7 | 64.9 ± 7.2 | 67.0 ± 5.9 | 67.0 ± 6.0 | <0.001 |
| Height | cm | 158.1 ± 8.3 | 158.6 ± 8.6 | 157.3 ± 8.4 | 157.7 ± 8.5 | <0.001 |
| Weight | kg | 57.7 ± 10.2 | 61.1 ± 10.9 | 61.2 ± 10.5 | 62.9 ± 10.7 | <0.001 |
| Body mass index | kg/m$^2$ | 23.0 ± 3.1 | 24.2 ± 3.4 | 24.7 ± 3.4 | 25.2 ± 3.5 | <0.001 |
| Systolic blood pressure | mmHg | 120 ± 12 | 150 ± 14 | 126 ± 9 | 150 ± 12 | <0.001 |
| Diastolic blood pressure | mmHg | 72 ± 8 | 88 ± 10 | 74 ± 8 | 85 ± 10 | <0.001 |
| Triglycerides | mg/dl | 121 ± 79 | 141 ± 102 | 132 ± 80 | 142 ± 93 | <0.001 |
| High-density lipoprotein | mg/dl | 61 ± 16 | 59 ± 16 | 56 ± 15 | 57 ± 15 | <0.001 |
| Low-density lipoprotein | mg/dl | 128 ± 31 | 130 ± 33 | 120 ± 28 | 123 ± 30 | <0.001 |
| HbA1c | % | 5.3 ± 0.7 | 5.5 ± 1.0 | 5.5 ± 0.8 | 5.6 ± 0.9 | <0.001 |
| eGFR | ml/min/1.73 m$^2$ | 58.0 ± 12.6 | 60.0 ± 15.6 | 55.2 ± 13.3 | 56.7 ± 14.7 | <0.001 |
| Low eGFR | %, yes | 82.8 | 74.5 | 82.6 | 76.8 | <0.001 |
| Proteinuria | %, + or more | 21.9 | 33.9 | 29.0 | 38.7 | <0.001 |
| Use of antihypertensive drugs | %, yes | 0 | 0 | 100 | 100 | - |
| Us of lipid-lowering drugs | %, yes | 11.0 | 7.9 | 31.1 | 27.0 | <0.001 |
| Diabetes treatment | %, yes | 4.3 | 3.9 | 11.9 | 13.9 | <0.001 |
| Cigarette smoking | %, yes | 12.4 | 14.2 | 11.0 | 10.9 | <0.001 |

Low eGFR; less than 60 ml/min/1.73 m$^2$

mortalities and CVD deaths in each hypertension treatment category by the total number of subjects in that subpopulation showed increased rates of both all-cause and CVD mortalities in these subjects, regardless of their CKD status, compared to subjects with untreated, normal BP (Table 3).

**Table 3. Number, all-cause mortality, and mortality due to cardiovascular disease (CVD) in subpopulations with and without chronic kidney disease (CKD) according to proteinuria (UP) and renal function (eGFR).**

| | | Normotensive Untreated | Hypertensive Untreated | Normotensive Treated | Hypertensive Treated |
|---|---|---|---|---|---|
| | UP | | | | |
| Number | (-) | 123199 | 35270 | 31915 | 24216 |
| | (+) | 4113 | 2597 | 2747 | 3147 |
| All-cause mortality | (-) | 1179 (0.96%) | 435 (1.23%) | 464 (1.45%) | 341 (1.41%) |
| | (+) | 79 (1.92%) | 78 (3.00%) | 83 (3.02%) | 86 (2.73%) |
| CVD mortality | (-) | 98 (0.08%) | 61 (0.17%) | 70 (0.22%) | 55 (0.23%) |
| | (+) | 6 (0.15%) | 19 (0.73%) | 15 (0.55%) | 22 (0.70%) |
| | Low eGFR | | | | |
| Number | (-) | 111868 | 32290 | 26892 | 21230 |
| | (+) | 15444 | 5577 | 7770 | 6133 |
| All-cause mortality | (-) | 1008 (0.90%) | 426 (1.32%) | 389 (1.45%) | 284 (1.34%) |
| | (+) | 250 (1.62%) | 87 (1.56%) | 158 (2.03%) | 143 (2.33%) |
| CVD mortality | (-) | 87 (0.08%) | 62 (0.19%) | 57 (0.21%) | 44 (0.21%) |
| | (+) | 17 (0.11%) | 18 (0.32%) | 28 (0.36%) | 33 (0.54%) |
| | CKD | | | | |
| Number | (-) | 108653 | 30341 | 25236 | 19356 |
| | (+) | 18659 | 7526 | 9426 | 8007 |
| All-cause mortality | (-) | 961 (0.88%) | 372 (1.23%) | 352 (1.39%) | 244 (1.26%) |
| | (+) | 297 (1.59%) | 141 (1.87%) | 195 (2.07%) | 183 (2.29%) |
| CVD mortality | (-) | 83 (0.08%) | 51 (0.17%) | 52 (0.21%) | 36 (0.19%) |
| | (+) | 21 (0.11%) | 29 (0.39%) | 33 (0.35%) | 41 (0.51%) |

The hazard ratio (HR) for all-cause mortality in each hypertensive status is shown in Fig 1. Subjects with and without proteinuria (Fig 1A) or low eGFR (Fig 1B) and those with and without CKD were analyzed separately, regardless of how the condition was defined (i.e., "overall CKD") (Fig 1C). Using the untreated normotensive subpopulation as a reference, the risk of all-cause mortality among patients with proteinuria was significantly increased among those with untreated hypertension (age- and sex-adjusted HR, 1.38 [95% confidence interval 1.01–1.89]; multivariable-adjusted HR, 1.45 [1.06–1.99]; Fig 1A). However, among subjects with low eGFR (Fig 1B) and those with overall CKD (Fig 1C), the HR for all-cause mortality did not differ significantly between subjects with untreated normotension and those with untreated hypertension, those who became normotensive during treatment, or those who remained hypertensive despite treatment. Overall, analyses for all-cause mortality risk did not identify hypertensive status as an independent risk factor in subjects diagnosed with CKD according to the presence of proteinuria or decreased renal function.

In contrast to all-cause mortality, relative to those for the subpopulation with untreated normal BP, the adjusted HRs for CVD mortality in subjects without proteinuria, without low eGFR, and without CKD showed significance (Fig 2). Moreover, HRs for CVD mortality in subjects with CKD were mostly significant and varied widely in each hypertension treatment category (filled circles in Fig 2): from 2.12 to 4.61 in subjects with proteinuria, from 2.27 to 3.08 in those with low eGFR, and from 2.30 to 3.28 in those with overall CKD. Summarizing the above, these analyses identified every hypertension treatment category as a risk factor for CVD mortality both in non-CKD and in CKD status, independent of well-known CVD risk factors, such as age, sex, BMI, and cigarette smoking.

## Discussion

When compared with the general population, patients with CKD are at increased risk for CVD mortality [4,29]. General-population-based observational studies have shown that, for a

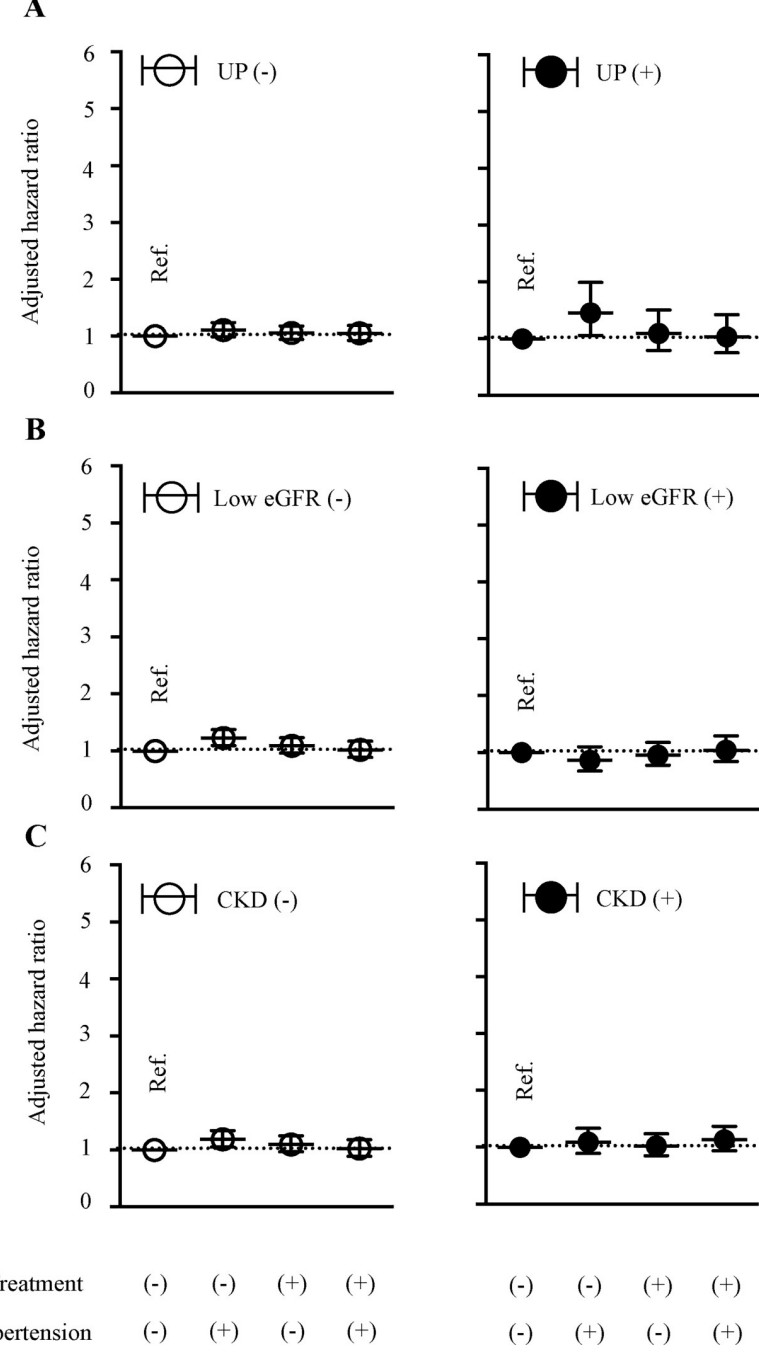

**Fig 1. Risk of all-cause mortality in each hypertension treatment category.** The multivariable-adjusted hazard ratio and 195% confidence interval for all-cause death in each hypertension treatment category are shown. The reference category is untreated, normal blood pressure. The subgroups reflect the presence (or absence) of proteinuria (A), reduced renal function (B), or chronic kidney disease (C). Adjusted factors for death were: age; sex; cigarette smoking; body mass index; proteinuria; levels of triglycerides, high-density lipoprotein, and low-density lipoprotein; use of lipid-lowering drugs; HbA1c; and treatment for diabetes.

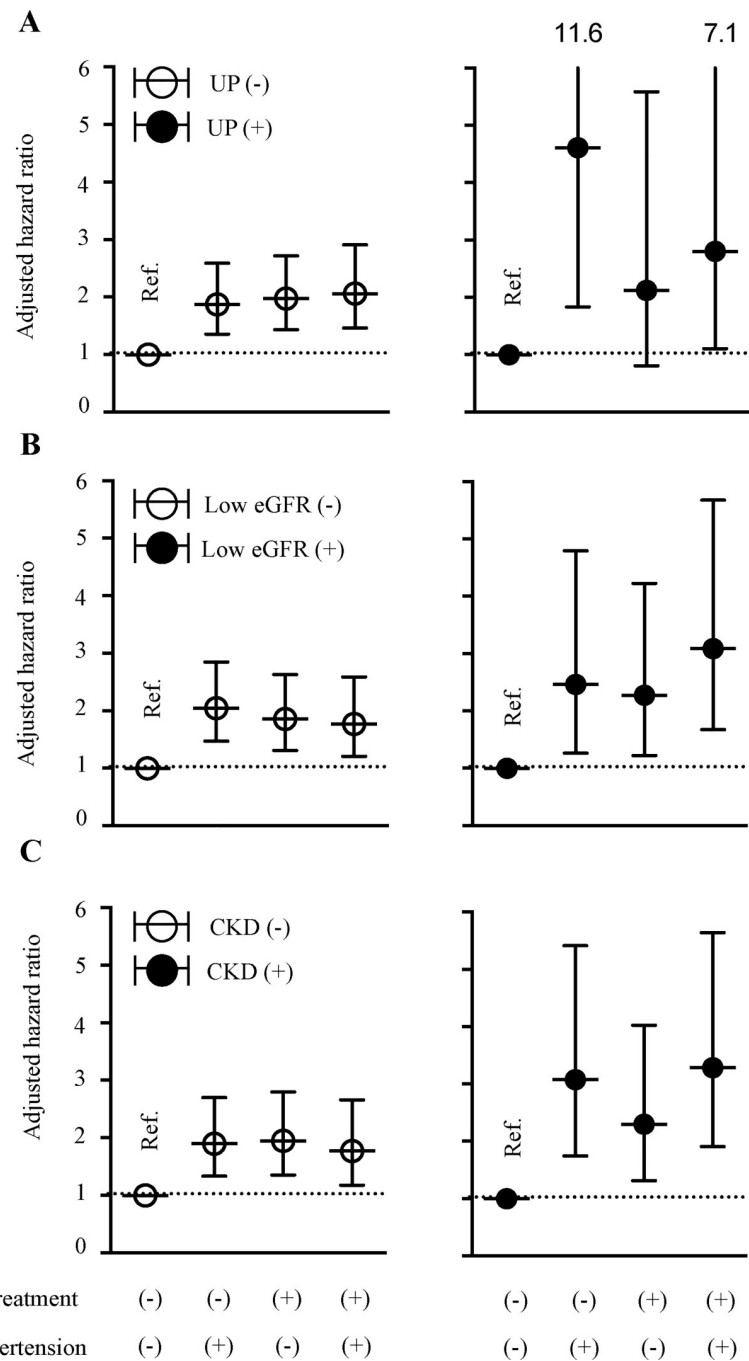

**Fig 2. Risk of all cardiovascular mortality in each hypertension treatment category.** The multivariable-adjusted hazard ratio and 95% confidence interval for cardiovascular mortality in each hypertensive treatment category are shown. The reference category is untreated, normal blood pressure. The subgroups reflect the presence (or absence) of proteinuria (A), reduced renal function (B), and chronic kidney disease (C). Adjusted factors for death were: age; sex; cigarette smoking; body mass index; proteinuria; levels of triglycerides, high-density lipoprotein, and low-density lipoprotein; use of lipid-lowering drugs; HbA1c; and treatment for diabetes.

given BP at baseline, CVD risk is higher in subjects who use antihypertensive medications than in those who do not [9–14]. Although the cohorts in the cited studies were well defined and rigorously followed, all of the cohorts were established decades ago (i.e., 1970 [9,10], 1991

through 1998 [11], 1967 through 1996 [13], and 1991 through 1996 [14]). In addition, their results were evaluated in the context of then-contemporary clinical practice. Therefore, most of the evidence [2,30–34] does not take into consideration the renal function of the participants, because the cohorts were also built without the conception of CKD from Kidney Disease: Improving Global Outcomes (KDIGO) in 2002 [6]. Because the timing and breadth of the recruitment might affect the characteristics of cohort subjects because of changes in clinical practice regarding the treatment of hypertension, use of a cohort with well-identified renal function and consistent baseline year data facilitates assessment of the relationship between BP control at baseline and CVD mortality in patients with CKD.

In this analysis, a general-population-based cohort established in 2008 was examined for which data regarding proteinuria, renal function, other CVD risk factors, and ICD-coded cause of death were available. In this cohort, age, risks, and proportions of patients at baseline year differed significantly between subjects with untreated normal BP and those in other hypertension treatment categories. Because ageing is associated with a decline in eGFR and increases in BP and the rate of positive proteinuria [35], the older age and lower eGFR seen in treated or hypertensive subjects are unsurprising (Table 1). Similarly, within the same hypertension treatment category, subjects with CKD tended to be older than those without CKD (Table 2).

Previous Japanese general-population–based cohorts established from 1980 through 1995 [2, 31–34] comprised 27.4% with untreated hypertension. Research and development in antihypertensive drugs has been remarkable. As examples, angiotensin-converting enzyme inhibitors began to be prescribed in 1982 to 1998 and angiotensin receptor blockers began in 1998 to 2012, and their market share in Japan continues to grow. Our investigation shows the rate of untreated hypertension was subjects with CKD (17.3%) than hose without CKD (16.5%), suggesting successful spread of anti-hypertensive drugs in recent years (Table 2).

A collaborative prospective meta-analysis of randomized trials to examine the cardiovascular effects of lowering BP in people with CKD according to renal function [36], which covers 152,290 participants, including 30,295 with eGFR <60 ml/min/1.73 m$^2$. Another [15] performed a meta-analysis of systematic searches of BP-lowering trials to examine the effects of a 10-mmHg reduction in systolic BP on the relative risk of major CVD in 30,766 participants from 18 cohorts. However, neither of these previous studies [15,36] directly showed the effect of BP reduction in patients with proteinuria. Though our study could not show any effect of interventional BP control on CVD mortality in patients with proteinuria, markedly high HR (4.61) in untreated hypertension category for CVD mortality in positive proteinuria population implies importance of BP control in such patients (Fig 2A).

The strength of this study was that it evaluated a large general population (i.e., more than 100,000 subjects) with available data regarding renal function, proteinuria, and CVD (i.e., stroke and cardiac events) mortality according to ICD-10 coding. These features allowed us to perform sub-analyses of the CKD and non-CKD subjects enrolled, showing that the risk of CVD in patients who remained hypertensive despite treatment differed between those with and without CKD, between those with and without proteinuria, and between those with and without low eGFR.

However, this study had several limitations. First, the database did not have details of the types of antihypertensive drugs used, such as renin-angiotensin-targeted drugs, which affect the incidence of CVD [37], because this information was self-reported and not obtained through medical records or claims. Second, the follow-up time (maximum, 4 years) was much shorter than in previous studies. Because this observational study could not show cause/result relationship, evidence for interventional benefit of BP control was not obtained.

Nevertheless, the results from a nationwide cohort could be one of representative demographics of controlling blood pressure and cardiovascular disease deaths when treating

patients with CKD in Japan in recent years. Even after current development and sufficient spread of anti-hypertensive drugs, preventing development of hypertension is preferable, because any hypertension treatment status comparing untreated normal blood pressure was a risk of CVD mortality at baseline year.

## Acknowledgments

This study would not have been possible without the generous support from the public health nurses of the Kokuho agency in each district. This study was supported in part by a grant for Development of Comprehensive Database of Chronic Kidney Disease, Research on Policy Planning and Evaluation from the Ministry of Health, Labor and Welfare of Japan.

## Author Contributions

**Conceptualization:** Kei Nagai, Kunihiro Yamagata, Kunitoshi Iseki, Koichi Asahi, Tsuyoshi Watanabe.

**Funding acquisition:** Kunihiro Yamagata.

**Investigation:** Kei Nagai, Kunihiro Yamagata, Kunitoshi Iseki, Koichi Asahi, Tsuyoshi Watanabe.

**Methodology:** Kei Nagai, Kunihiro Yamagata, Kunitoshi Iseki, Koichi Asahi, Tsuyoshi Watanabe.

**Supervision:** Kunihiro Yamagata, Kunitoshi Iseki, Tsuyoshi Watanabe.

**Visualization:** Kei Nagai.

**Writing – original draft:** Kei Nagai.

**Writing – review & editing:** Kunihiro Yamagata, Kunitoshi Iseki, Toshiki Moriyama, Kazuhiko Tsuruya, Shouichi Fujimoto, Ichiei Narita, Tsuneo Konta, Masahide Kondo, Masato Kasahara, Yugo Shibagaki, Koichi Asahi, Tsuyoshi Watanabe.

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
