## [Decision Letter · Decision Letter 0]

10 Oct 2019

PONE-D-19-26343

Antihypertensive treatment and risk of cardiovascular mortality in patients with chronic kidney disease diagnosed based on the presence of proteinuria and renal function: A large longitudinal study in Japan

PLOS ONE

Dear Dr. Yamagata,

Thank you for submitting your manuscript to PLOS ONE. After careful consideration, we feel that it has merit but does not fully meet PLOS ONE’s publication criteria as it currently stands. Therefore, we invite you to submit a revised version of the manuscript that addresses the points raised during the review process.

The journal office told me that the PLOS ONE journal strongly discourages the unnecessary division of work into separate manuscripts. Each submission must be written as an independent unit and should not rely on any work that has not already been accepted for publication. As an editor, I may comment on overlap between related submissions and advise authors to combine their submissions into a single manuscript. To learn more about PLOS ONE's policies on related studies, visit: http://journals.plos.org/plosone/s/ethical-publishing-practice#loc-submission-and-publication-of-related-studies

Therefore, I recommend the authors to describe the difference between this study and the previous study about the importance of transient dipstick-proteinuria on mortality you have published recently in your reply.

We would appreciate receiving your revised manuscript by Nov 24 2019 11:59PM. To enhance the reproducibility of your results, we recommend that if applicable you deposit your laboratory protocols in protocols.io, where a protocol can be assigned its own identifier (DOI) such that it can be cited independently in the future. For instructions see: http://journals.plos.org/plosone/s/submission-guidelines#loc-laboratory-protocols

We look forward to receiving your revised manuscript.

Kind regards,

Kojiro Nagai

Academic Editor

PLOS ONE

Journal Requirements:

Reviewers' comments:

Reviewer's Responses to Questions

**Comments to the Author**

1. Is the manuscript technically sound, and do the data support the conclusions?

Reviewer #1: No

Reviewer #2: Yes

2. Has the statistical analysis been performed appropriately and rigorously? 

Reviewer #1: No

Reviewer #2: Yes

3. Have the authors made all data underlying the findings in their manuscript fully available?

Reviewer #1: Yes

Reviewer #2: Yes

4. Is the manuscript presented in an intelligible fashion and written in standard English?

Reviewer #1: Yes

Reviewer #2: Yes

5. Review Comments to the Author

Reviewer #1: In the present study, Kei Nagai et al. examined mortality due to cardiovascular disease (CVD) among subjects who underwent annual health check in Japan. Study population was approx. 220 thousand and the follow up period was 4 years. The health check participants were separated into chronic kidney disease (CKD) and non-CKD subjects. They were also divided by their blood pressure/hypertension treatment categories. The following was one of the major findings: in CKD patients, compared with CVD mortality in patients with untreated normal blood pressure, the multivariable adjusted hazard ratio was 3.08 (95% confidence interval: 1.75–5.41) for those with untreated hypertension and 2.30 (1.31–4.03) for those who became normotensive after treatment, and 3.28 (1.91–5.64) for those who remained hypertensive despite treatment. They concluded, among subjects with CKD, patients who become normotensive during treatment have a lower risk of CVD than those who remain hypertensive, suggesting the effectiveness of BP reduction in preventing CVD death in CKD patients. Unfortunately, findings presented in this work do not support this conclusion.

[Major Points]

1) Since CVD mortality in CKD patients with normotensive/hypertension treated category (0.35%) was lower than that with hypertensive/untreated category (0.39%), the authors interpreted that the findings support treatment of hypertension in CKD patients to reduce CVD death. However, this is an observational study and cause/result relationship and speculation for prospective interventional benefits cannot be obtained, since patients in these 2 categories may have different basal backgrounds. For instance, patients with untreated hypertension may have lower socioeconomical conditions and may be less concerned about their health. Normotensive patients under hypertension treatment may have a better chance to be diagnosed other disorders during their periodical hospital visit and receive better care. There is no guarantee that untreated subjects remain untreated for 4 years.

2) Indeed, CVD mortality in non-CKD subjects were 0.21% for normotensive/hypertension treated category and 0.17% for hypertensive/untreated category (Table 3). Furthermore, CVD mortality in overall population (including CKD and non-CKD) were 0.27% for normotensive/treated category and 0.22% for hypertensive/untreated category (by reviewer’s calculation from Table 3). These findings prove that authors’ logic is inconsistent to established effects of hypertension treatment to reduce CVD death in CKD and non-CKD subjects.

3) As the authors described, previous studies reported CVD mortality among subjects with reduced renal function. The present study is distinct in that not only reduced renal function but also proteinuria was analyzed.

4) Median follow up period should be described in the abstract.

5) CVD mortality in CKD and non-CKD subjects should be described in the abstract.

[Minor Points]

1) A term “hypertensive category” is often used in the manuscript but it implies the extent of hypertension: for instance, 140, 160 and 180 mmHg. Other term, such as hypertension treatment categories, should be appropriate.

2) Line 138: >5 years should be replaced by 5.5 years.

Reviewer #2: Nagai et al reported the association between antihypertensive treatment and CVD mortality in patients with CKD who received annual health checkups. This study is a longitudinal general-population cohort with a large number of subject and showed the clear results of an increased mortality in those with hypertension or with hypertensive treatment, thus the data is of clinical importance. However, this manuscript has some major problems.

1)In Fig2, authors showed hazard ratio of hypertensive group or treated groups, referencing untreated normal blood pressure. Authors conclusion is the effectiveness of BP reduction in preventing CVD in CKD patients, however, in this manuscript, it is unclear whether hazard ration between untreated hypertensive people, treated normal BP people and treated hypertensive people in CKD is statistically different. Authors should clearly show this point.

2)In Fig2, authors showed hazard ration of CVD mortality in non-CKD people. These data revealed the CVD risk of hypertension, however did not show the effectiveness of hypertensive treatment for the prevention of CVD risk in non-CKD people. Is BP control not effective for the CVD reduction in non-CKD hypertensive subjects in this study? Authors should describe the results although this manuscript targets CKD patients.

3) Table3 showed CVD mortality in those with CKD and without CKD. In conclusions, authors described “…CVD mortality risk seems to be higher in CKD patients than in non-CKD patients.”. Authors should show statistical significance.

6. PLOS authors have the option to publish the peer review history of their article (what does this mean?). If published, this will include your full peer review and any attached files.

Reviewer #1: No

Reviewer #2: No

---

## [Author Response · Author response to Decision Letter 0]

30 Oct 2019

To the reviewers;

Reviewer #1: 

[Major points]

1-1) Since CVD mortality in CKD patients with normotensive/hypertension treated category (0.35%) was lower than that with hypertensive/untreated category (0.39%), the authors interpreted that the findings support treatment of hypertension in CKD patients to reduce CVD death. However, this is an observational study and cause/result relationship and speculation for prospective interventional benefits cannot be obtained, since patients in these 2 categories may have different basal backgrounds. For instance, patients with untreated hypertension may have lower socioeconomical conditions and may be less concerned about their health. Normotensive patients under hypertension treatment may have a better chance to be diagnosed other disorders during their periodical hospital visit and receive better care. There is no guarantee that untreated subjects remain untreated for 4 years.

---Thank you for your comment and we recognize this observational study cannot show the interventional effect of BP control on CVD mortality and hard to discuss magnitude of HRs due to unadjustable factors among hypertension treatment categories before baseline year such as socioeconomical status and hospital visiting. Therefore, we would like to add limitation in the second last paragraph of the discussion and deleted description to compare HRs between two hypertension treatment categories other than reference group.

1-2) CVD mortality in non-CKD subjects were 0.21% for normotensive/hypertension treated category and 0.17% for hypertensive/untreated category (Table 3). Furthermore, CVD mortality in overall population (including CKD and non-CKD) were 0.27% for normotensive/treated category and 0.22% for hypertensive/untreated category (by reviewer's calculation from Table 3). These findings prove that authors' logic is inconsistent to established effects of hypertension treatment to reduce CVD death in CKD and non-CKD subjects.

---Thank you for your thoughtful comment. Related to your 1st major point, we recognized that it is hazardous to compare HRs each other between two hypertension treatment categories other than reference group (i.e. untreated normal BP). We would like to correct the conclusions from “effect of hypertension treatment on mortality” into “current demographics of hypertension treatment in Japan”. Please approve to mix up discussion and conclusion sections in order to keep context fluency (PLoS One editorial approves it by the author instruction).

1-3) As the authors described, previous studies reported CVD mortality among subjects with reduced renal function. The present study is distinct in that not only reduced renal function but also proteinuria was analyzed.

---Thank you for your suggestion, we highlighted strong point of our study to deal with proteinuria, in the fourth paragraph of the revised discussion. 

1-4) Median follow up period should be described in the abstract.

---Thank you for your comment. We added follow-up period in the revised abstract and method section.

1-5) CVD mortality in CKD and non-CKD subjects should be described in the abstract.

---Thank you for your comment. We added description regarding CVD mortality in the revised abstract.

[Minor points]

2-1) A term "hypertensive category" is often used in the manuscript but it implies the extent of hypertension: for instance, 140, 160 and 180 mmHg. Other term, such as hypertension treatment categories, should be appropriate.

---Thank you so much for your suggestion and we agree with you. We used “hypertension treatment category” throughout this manuscript.

2-2) Line 138: >5 years should be replaced by 5.5 years.

---Thank you for comment. Please make sure it in the revised version.

Reviewer #2: 

1) In Fig2, authors showed hazard ratio of hypertensive group or treated groups, referencing untreated normal blood pressure. Authors conclusion is the effectiveness of BP reduction in preventing CVD in CKD patients, however, in this manuscript, it is unclear whether hazard ration between untreated hypertensive people, treated normal BP people and treated hypertensive people in CKD is statistically different. Authors should clearly show this point.

---Thank you for your thoughtful comment. As another reviewer and you pointed out, we recognized that it is hard to compare HRs each other between two hypertension treatment categories (ie. untreated hypertension vs treated normal BP), other than reference group (untreated normal BP). We would like to correct the conclusions from “effect of hypertension treatment” into “current demographics of hypertension treatment in Japan”. Please approve to mix up discussion and conclusion sections in order to keep context fluency (PLoS One editorial approves it by the author instruction).

2) In Fig2, authors showed hazard ration of CVD mortality in non-CKD people. These data revealed the CVD risk of hypertension, however did not show the effectiveness of hypertensive treatment for the prevention of CVD risk in non-CKD people. Is BP control not effective for the CVD reduction in non-CKD hypertensive subjects in this study? Authors should describe the results although this manuscript targets CKD patients.

---Related to your 1st point, we should not have compared HRs between untreated hypertension and treated normal BP, among CKD subjects as well as among non-CKD. We appropriately deleted description regarding effectiveness of BP control and just touched on the previous works in CKD cohort and implication of marked CVD mortality risk in patients with positive proteinuria in the fourth paragraph of the revised manuscript. 

3) Table3 showed CVD mortality in those with CKD and without CKD. In conclusions, authors described "...CVD mortality risk seems to be higher in CKD patients than in non-CKD patients.". Authors should show statistical significance.

---Thank you for your thoughtful comment. Related to your former comments, we revised conclusion, because it is difficult to compare HRs among hypertension treatment categories. Please make sure the last paragraph of the revised discussion section. 

To the editorial;

We deeply recognized your journal office has been afraid of overlapping contents in our current and previous manuscripts. In advance of launching this cohort study using health checkups before 2008, our committee determined “one project, one publication-policy”. It means that each researcher has been analyzed based on the standard analytical file exactly for the purposes of testing his or her original scientific interest - in chronic kidney disease (CKD), clinical relevance of dipstick proteinuria for general population and distribution of hypertension and its treatment in Japan, etc. – with prior consultation. And then, the steering committee (SC) has supervised and carefully checked possibility of overlapping each other before submitting manuscript. In other words, our policy secures independence of each project and helps to avoid any double publications.

This project focused on demographics of hypertension and its treatment, while the previous one (PLoS One 14: e0223005) focused on sequential results of CKD diagnosis (ie. renal function and proteinuria) and its mortality risk. Therefore, we would like to propose obvious independence between these two projects.

Please refer our position paper shown below and you can see the list (attached in the end of this letter) and find “Antihypertensive treatment and risk of CVD death (Kunihiro Yamagata)”. Unfortunately, though the most updated list of our projects including “Cause-specific mortality in the general population with transient dipstick-proteinuria (PLoS One 14: e0223005)” is not published yet, SC authorized “Cause-specific mortality...” as independent project of “Antihypertensive treatment and risk of CVD death”.

*Clin Exp Nephrol. 2017 Dec;21(6):978-985. doi: 10.1007/s10157-017-1392-y. Epub 2017 Mar 3.

Mortality risk among screened subjects of the specific health check and guidance program in Japan 2008-2012.

Iseki K, Asahi K, Yamagata K, Fujimoto S, Tsuruya K, Narita I, Konta T, Kasahara M, Shibagaki Y, Yoshida H, Moriyama T, Kondo M, Iseki C, Watanabe T; “Design of the comprehensive health care system for chronic kidney disease (CKD) based on the individual risk assessment by Specific Health Check”.

---

## [Decision Letter · Decision Letter 1]

14 Nov 2019

Antihypertensive treatment and risk of cardiovascular mortality in patients with chronic kidney disease diagnosed based on the presence of proteinuria and renal function: A large longitudinal study in Japan

PONE-D-19-26343R1

Dear Dr. Yamagata,

We are pleased to inform you that your manuscript has been judged scientifically suitable for publication and will be formally accepted for publication once it complies with all outstanding technical requirements.

With kind regards,

Kojiro Nagai

Academic Editor

PLOS ONE

Additional Editor Comments (optional):

Reviewers' comments:

Reviewer's Responses to Questions

**Comments to the Author**

1. If the authors have adequately addressed your comments raised in a previous round of review and you feel that this manuscript is now acceptable for publication, you may indicate that here to bypass the “Comments to the Author” section, enter your conflict of interest statement in the “Confidential to Editor” section, and submit your "Accept" recommendation.

Reviewer #1: All comments have been addressed

Reviewer #2: All comments have been addressed

2. Is the manuscript technically sound, and do the data support the conclusions?

Reviewer #1: (No Response)

Reviewer #2: Partly

3. Has the statistical analysis been performed appropriately and rigorously? 

Reviewer #1: (No Response)

Reviewer #2: Yes

4. Have the authors made all data underlying the findings in their manuscript fully available?

Reviewer #1: (No Response)

Reviewer #2: Yes

5. Is the manuscript presented in an intelligible fashion and written in standard English?

Reviewer #1: (No Response)

Reviewer #2: Yes

6. Review Comments to the Author

Reviewer #1: (No Response)

Reviewer #2: (No Response)

7. PLOS authors have the option to publish the peer review history of their article (what does this mean?). If published, this will include your full peer review and any attached files.

Reviewer #1: No

Reviewer #2: No

---

## [Editor Report · Acceptance letter]

22 Nov 2019

PONE-D-19-26343R1 

Antihypertensive treatment and risk of cardiovascular mortality in patients with chronic kidney disease diagnosed based on the presence of proteinuria and renal function: A large longitudinal study in Japan 

Dear Dr. Yamagata:

I am pleased to inform you that your manuscript has been deemed suitable for publication in PLOS ONE. Congratulations! Your manuscript is now with our production department. 

With kind regards,

on behalf of

Dr. Kojiro Nagai 

Academic Editor

PLOS ONE